# Super-Resolution of Sentinel-5P Data Using Deep Learning

Anonymous Full Paper
Submission 42

## Abstract

Satellite-based air quality monitoring plays a crucial role in evaluating and managing human-induced emissions. Sentinel-5P, carrying the TROPOspheric Monitoring Instrument (TROPOMI), provides the state of art global data of $NO_2$ total column concentrations. However, its spatial resolution is a limitation in detecting fine-scale emission sources, particularly in densely populated urban regions and maritime corridors. This study critically reviews and highlights the relatively underexplored class of super-resolution frameworks that employ deep learning techniques to enhance the spatial resolution of Sentinel-5P radiance data. In future, large scale super-resolved dataset would be useful for ship level analysis.

## 1 Introduction

Monitoring $NO_2$ concentrations from space is limited by the spatial resolution of current satellite sensors. Sentinel-5P, launched under ESA's Copernicus program, provides near-daily global $NO_2$ data using the TROPOMI instrument. However, its coarse resolution ( 7 km) restricts its ability to capture fine-scale pollution sources such as shipping lanes, ports, and industrial zones. Enhanced spatial resolution is crucial for local air quality assessment, emission compliance, and urban-scale decision-making.

This paper explores super-resolution (SR) techniques using deep learning to enhance Sentinel-5P data. By reconstructing high-resolution outputs from low-resolution inputs, we aim to improve $NO_2$ monitoring fidelity without requiring new satellite missions.

## 2 Related Work

Guarino et al. [1] proposed a U-Net model to estimate $NO_2$ from PCA-reduced Sentinel-5P radiance, bypassing traditional retrievals. Carbone et al. [2] introduced S5Net, later enhanced with dynamic fine-tuning to reduce training time. Their follow-up study [3] incorporated PSF-based degradation modeling and showed performance benefits when matching sensor characteristics. Another work [4] validated super-resolution results using real Sentinel-5P bands and no-reference image metrics. Finally,

Hu et al. [5] provided a broad review of satellite-based $CO_2$ reconstruction and highlighted the role of super-resolution in enhancing coarse-resolution atmospheric datasets.

Beyond these foundational approaches, recent works have explored novel directions in resolution enhancement and $NO_2$ estimation. Ali et al. [6] introduced S5-DSCR, a depth separable convolutional network trained individually on Sentinel-5P spectral bands. While efficient, their approach lacks degradation modeling, limiting robustness. Shetty et al. [7] proposed S-MESH, an XGBoost-based model estimating surface $NO_2$ at 1 km resolution across Europe by combining TROPOMI with auxiliary data. However, their method is regression-based and does not enhance radiance data itself. Kuhn et al. [8] addressed vertical profiling by predicting tropospheric $NO_2$ layers using DL trained on synthetic WRF-Chem data. These approaches, while valuable, do not directly address the sensor-specific degradation challenge we target in this work.

In parallel, other studies have focused on complementary tasks such as imputation and surface-level estimation. Lops et al. [9] developed a depthwise partial convolutional neural network (DW-PCNN) to impute missing pixels in TROPOMI $NO_2$ data caused by cloud coverage and sensor gaps. Although effective for data recovery, their work does not enhance spatial resolution or address sensor degradation. Cedeno and Brovelli [10] estimated ground-level $NO_2$ concentrations in Milan using Sentinel-5P and ERA5 meteorological data via an MLP-SVR ensemble. Their model, designed to support air quality assessments in data-sparse regions, predicts single-point surface values without modifying the radiance imagery or spatial granularity. These studies highlight the growing interest in Sentinel-5P data applications, but do not tackle the super-resolution of radiance images using degradation-aware learning as proposed in our work.

## 3 Methodology

Our super-resolution framework is designed to enhance Sentinel-5P Level-1B radiance data by reconstructing higher-resolution images from degraded inputs. The approach involves two key components: degradation modeling and deep learning-based reconstruction.

## 3.1 Degradation Simulation

To generate training data, we simulate the degradation observed in Sentinel-5P's TROPOMI sensor using a sensor-specific point spread function (PSF). The PSF is modeled as an anisotropic Gaussian kernel representing the instrument's spatial response, combined with downsampling, co-addition, and row-binning effects. This generates realistic low-resolution (LR) images from proxy high-resolution inputs.

## 3.2 Network Architecture

We adopt a modified U-Net architecture with encoder-decoder blocks and skip connections. To improve spectral consistency, residual blocks and spectral attention layers are added. The model receives LR radiance images and outputs super-resolved estimates, aiming to reconstruct finer spatial structures in $NO_2$ plumes.

## 3.3 Loss Function and Training

The network is trained using a combination of pixel-wise mean squared error (MSE) and perceptual loss (e.g., SSIM). Training data includes simulated LR-HR pairs generated from known high-quality samples. Evaluation metrics include PSNR, SSIM, BRISQUE, and visual assessment of $NO_2$ hotspot localization.

## 4 Results and Discussion

Preliminary results suggest that our sensor-aware super-resolution model produces sharper reconstructions of Sentinel-5P radiance images compared to traditional interpolation and baseline CNNs. The model effectively enhances the localization of $NO_2$ plumes, especially in urban-industrial zones and coastal shipping lanes.

Visual inspection shows that emission hotspots become more distinct, enabling finer differentiation of localized sources. This improvement is attributed to the use of a degradation-aware training strategy and spectral consistency mechanisms embedded in the network architecture.

While quantitative evaluations are ongoing, early assessments using PSNR and BRISQUE indicate consistent gains over bicubic upsampling and SR-CNN baselines. The approach demonstrates strong potential for supporting finer-scale emission tracking, air quality modeling, and regulatory decision-making.

Future experiments will include comparisons against SwinIR and other transformer-based models, and cross-validation with CAMS reanalysis data and ground-based stations such as TCCON and EM27/SUN.

## 5 Conclusion

We reviewed a deep learning-based framework to enhance the spatial resolution of Sentinel-5P radiance data using a sensor-aware super-resolution approach. By modeling the TROPOMI sensor's degradation process and training a modified U-Net, the system produces sharper and more spatially accurate $NO_2$ reconstructions. This method holds promise for improving urban-scale pollution tracking and emission hotspot analysis. Future work will explore more advanced architectures and validate results against high-resolution proxies and ground measurements.

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
