# OpenReview forum: "Super-Resolution of Sentinel-5P Data Using Deep Learning"
_NLDL.org/2026/Abstracts_Track — NLDL 2026 Abstracts_

### Official Review · Reviewer_L8SD · 2025-10-27

**Soundness:** 3
**Correctness:** 3
**Rating:** 4
**Confidence:** 4

**Summary:**

The work aims to generate super-resolution air quality measurements from Sentinel-5P satellite observations in order to enable detection of pollution sources. High-resolution training data is synthetically degraded using a physics-based simulator to mimic satellite measurements. The focus is on solving a specific application challenge rather than methodological innovation, but absolutely carries potential for interesting discussions during the conference.

**Strengths:**

- The work is clearly positioned within related work.
- Combines degradation modelling with deep learning based reconstruction.

**Weaknesses:**

- Relevant training data should be presented early in the data – what are the available “true” high-resolution emission data.
- Relies on simulated degradation; the realism and limitations of this approach should be discussed.
- Given the existence of a physics based simulator of the instrument, it should be discussed why a similar simulator could not up-scale the images.

---

### Official Review · Reviewer_5Qpg · 2025-11-02

**Soundness:** 2
**Correctness:** 2
**Rating:** 4
**Confidence:** 1

**Summary:**

The abstract presents an approach to enhancing Sential-5p level-1B radiance data. The approach is based on 2 components: degradation modelling and reconstruction based on a modified U-net model.

**Strengths:**

- The motivation behind the goal of enhancing spatial resolution of Sentinal-5P data is clear and a good use-case for deep learning models.
- The section on related work is thorough.

**Weaknesses:**

- The abstract does not present concrete results, but only describes what preliminary findings indicate. Including clearer presentation of these preliminary results would greatly improve the quality.
- The description of the proposed approach could be made more detailed.
- Providing a clearer motivation for why this particular approach or design was chosen over alternative approaches would strengthen the justification for the work.

---

### Official Review · Reviewer_fmMf · 2025-11-03

**Soundness:** 2
**Correctness:** 3
**Rating:** 4
**Confidence:** 3

**Summary:**

This study critically reviews and highlights the relatively underexplored class of super-resolution frameworks that use deep learning techniques to enhance the spatial resolution of Sentinel-5P radiance data. They explore super-resolution approaches to come over the problem of coarse resolution which hinders the ability to detect fine scale pollution sources.

**Strengths:**

* Well written and clear
* Detailed literature review

**Weaknesses:**

* An illustration of the approach would make it much easier to follow the details, specifically there is space left
* No quantitative results

---

### Decision · Program_Chairs · 2025-11-05

**Decision:**

Accept

**Comment:**

The abstract is of interest to the community and should be presented at the conference.